# Text Classification with Born's Rule

**Emanuele Guidotti**
Institute of Financial Analysis
University of Neuchâtel, Switzerland
emanuele.guidotti@unine.ch

**Alfio Ferrara**
Department of Computer Science and Data Science Research Center
University of Milan, Italy
alfio.ferrara@unimi.it

## Abstract

This paper presents a text classification algorithm inspired by the notion of superposition of states in quantum physics. By regarding text as a superposition of words, we derive the wave function of a document and we compute the transition probability of the document to a target class according to Born's rule. Two complementary implementations are presented. In the first one, wave functions are calculated explicitly. The second implementation embeds the classifier in a neural network architecture. Through analysis of three benchmark datasets, we illustrate several aspects of the proposed method, such as classification performance, explainability, and computational efficiency. These ideas are also applicable to non-textual data.

## 1 Introduction

In quantum physics, the superposition principle is the idea that a system can be in multiple states at the same time (e.g., a cat is simultaneously alive and dead in Schrödinger's thought experiment [22]). Here we show that text can be formally treated as a quantum system in a superposition of words. Our superposition-of-words model is a bag-of-words model where each word is treated as a quantum state. By exploiting this representation, we develop a supervised classifier based on key postulates of quantum mechanics, namely the Born rule [4].

The Born rule provides a link between the mathematical formalism of quantum theory and experiment, and as such is almost single-handedly responsible for practically all predictions of quantum physics.[1] In this paper, we represent documents and classes as quantum objects and we compute the probability of a document to collapse in a target class by applying the Born rule.

The paper is structured as follows. Section 2 presents the motivation of this work. Section 3 introduces the notation and some preliminary notions in quantum mechanics. Section 4 develops our classification algorithm. Section 5 embeds the classifier in a neural network architecture. Section 6 presents our empirical results. Finally, Section 7 gives our concluding remarks and discusses extensions to semantics and non-textual data.

To simplify extensions to this work, we implement our classification algorithm in scikit-learn [17] and we embed the classifier in a neural network architecture using pytorch [16]. All the code is available at https://github.com/eguidotti/bornrule.

---

[1]We refer the reader to [14] for an introduction to the Born rule.

## 2 Motivation

In its most general form, the no-free-lunch theorem [23] implies that only prior knowledge makes it possible to generalize from the training examples to novel test examples. As quantum mechanics represents our understanding of nature at the deepest level [13], we wonder whether quantum-inspired machine learning may incorporate a fundamental form of prior knowledge. Our hope is that such knowledge results in more efficient algorithms, with better generalization ability and shorter computational times. Such algorithms would be also beneficial in terms of explainability and interpretability [8], as their interpretation is immediately inherited from the physical model [18].[2]

This work should not be confused with quantum computing, which harnesses the properties of quantum states to perform calculations [1], or with quantum machine learning [3], which explores how to devise and implement quantum software that could enable machine learning that is faster than that of classical computers [6]. We refer to our work as quantum-inspired machine learning, in that we use quantum theory to derive machine learning algorithms that can run on classical computers. Although this is not the first time quantum theory is applied to machine learning (see e.g., [15]), this is, to our knowledge, the first step to develop a new general-purpose algorithm for supervised classification based on key postulates of quantum mechanics, namely the Born rule.

Here we focus on text classification, as the analogy with quantum systems is straightforward.[3] However, the formalism we present is general and the implementations we derive can be used in practice as general-purpose classifiers.

## 3 Notation

Let $\mathbf{x}$ be a feature vector with elements $x_j$ for $j = 1, ..., J$. Let $\mathbf{y}$ be a probability vector with elements $y_k \geq 0$ for $k = 1, ..., K$, and such that $\sum_k y_k = 1$. Our goal is to learn a function $g$ such that $\mathbf{y} = g(\mathbf{x})$. Then, given a test instance $\mathbf{x}'$, we predict the probabilities $\mathbf{y}' = g(\mathbf{x}')$ and select the class $k^* = \mathrm{argmax}_k \, y'_k$ for classification.

We start by giving some preliminary notions in quantum mechanics.

**Wave function.**    In quantum physics, a system is regarded as a superposition of states $|s\rangle$ and, using Dirac's notation [7], it is represented by a wave function $|\psi\rangle$:

$$|\psi\rangle = \sum_s \psi_s |s\rangle \quad \text{with} \quad \psi_s \in \mathbb{C}. \tag{1}$$

**Born rule.**    In the Copenhagen interpretation, the modulus squared of the inner product is interpreted as the (unnormalized) probability of the wave function $|\psi\rangle$ collapsing to a new wave function $|\varphi\rangle$:

$$P(\psi \to \varphi) = |\langle\varphi|\psi\rangle|^2 = \left|\sum_s \bar{\varphi}_s \psi_s\right|^2, \tag{2}$$

where $\bar{\varphi}_s$ denotes the complex conjugate of $\varphi_s$. This is known as the Born rule, and it is one of the fundamental postulates of quantum mechanics.

**Wave coefficients.**    From (1) and (2), we notice that the coefficient $|\psi_s|^2$ represents the (unnormalized) probability of the wave function $|\psi\rangle$ to collapse in state $|s\rangle$:

$$P(\psi \to s) = |\langle s|\psi\rangle|^2 = |\psi_s|^2. \tag{3}$$

## 4 Classification Algorithm

Let the feature vector $\mathbf{x}$ contain only non-negative elements such that $x_j \geq 0$ for all $j$ (e.g., word counts or tf-idf weights). We regard $x_j$ as the (unnormalized) probability of the data instance (e.g., document) to collapse in the $j$-th feature (e.g., word). We represent the $j$-th feature as a quantum state

---

[2]We refer the reader to [19] for an introduction to quantum mechanics and its interpretation.

[3]We refer the reader to [2, 5, 21, 27, 28] for quantum language models.

$|j\rangle$, and we represent the data instance with a superposition of states $|\psi\rangle = \sum_j \psi_j |j\rangle$. According to (3) we have $x_j = P(\psi \to j) = |\psi_j|^2$, which implies the natural choice $\psi_j = \sqrt{x_j}$.

$$|\psi\rangle = \sum_j \psi_j |j\rangle = \sum_j \sqrt{x_j} |j\rangle \tag{4}$$

In a similar way, we represent the $k$-th class with a wave function $|\varphi^{(k)}\rangle = \sum_j \varphi_j^{(k)} |j\rangle$, and we obtain the coefficients $\varphi_j^{(k)}$ by setting the transition probability from $|\varphi^{(k)}\rangle$ to $|j\rangle$ equal to the conditional probability of feature $j$ given class $k$, which we write as $P_{j|k}$. According to (3) we have $P_{j|k} = P(\varphi^{(k)} \to j) = |\varphi_j^{(k)}|^2$, which implies the natural choice $\varphi_j^{(k)} = \sqrt{P_{j|k}}$.

$$|\varphi^{(k)}\rangle = \sum_j \varphi_j^{(k)} |j\rangle = \sum_j \sqrt{P_{j|k}} |j\rangle \tag{5}$$

Finally, we obtain the classification probability $y_k$ by computing the probability of $|\psi\rangle$ to collapse in $|\varphi^{(k)}\rangle$. By substituting (4) and (5) in (2), the unnormalized probabilities are:

$$u_k = P(\psi \to \varphi^{(k)}) = |\langle \varphi^{(k)} | \psi \rangle|^2 = \left| \sum_j \bar{\varphi}_j^{(k)} \psi_j \right|^2 = \left( \sum_j \sqrt{P_{j|k} x_j} \right)^2, \tag{6}$$

and the normalized probabilities are $y_k = u_k / \sum_k u_k$.

## 4.1 Training

To obtain the conditional probability $P_{j|k}$ in (6) we proceed as follows. Given a training set $\{(\mathbf{x}^{(n)}, \mathbf{y}^{(n)})\}_{n=1,...,N}$, we normalize each feature vector $\mathbf{x}^{(n)}$ such that it sums up to 1:

$$z_j^{(n)} = \frac{x_j^{(n)}}{\sum_{j'} x_{j'}^{(n)}}. \tag{7}$$

Then, we compute the conditional probability $P_{j|k}$ from the (unnormalized) joint probability $P_{jk}$:

$$P_{jk} = \sum_n z_j^{(n)} y_k^{(n)}, \quad P_{j|k} = \frac{P_{jk}}{\sum_{j'} P_{j'k}}. \tag{8}$$

## 4.2 Regularization

We observe that if $P_{j|k}$ is constant for $k = 1, ..., K$, then the $j$-th addend increases the summation in (6) by the same value for all classes. As $u_k$ is a monotonic transformation of the summation, the $j$-th addend does not alter the ranking of the probabilities $u_k$, thus being irrelevant for the final classification $k^* = \arg\max_k y_k = \arg\max_k u_k$. To regularize the predictions, we shrink the contribution of irrelevant addends towards zero by re-weighting the summation in (6).

Let us rewrite, for ease of notation, the probabilities $P_{j|k}$ as some weights $W_{jk} \geq 0$. Then, $j$ is irrelevant if $W_{jk}$ is constant for all $k$. We normalize $W_{jk}$ such that the weights of the classes $k$ sum up to 1 for each $j$, that is:

$$W_{k|j} = \frac{W_{jk}}{\sum_{k'} W_{jk'}}. \tag{9}$$

An irrelevant $j$ maximizes the entropy $\mathcal{H}_j = - \sum_k W_{k|j} ln(W_{k|j})$, as $W_{k|j}$ is uniformly distributed across the classes $k$. Thus, we introduce the following weights that range between 0 (irrelevant $j$ with maximum entropy) and 1 (relevant $j$ with null entropy):

$$H_j = 1 - \frac{\mathcal{H}_j}{\mathcal{H}_{max}} = 1 + \frac{\sum_k W_{k|j} \ln(W_{k|j})}{\ln(\sum_k 1)}. \tag{10}$$

Finally, we use $H_j$ in (10) to re-weight the summation in (6), which becomes:

$$u_k = \left( \sum_j H_j \sqrt{P_{j|k} x_j} \right)^2. \tag{11}$$

### 4.3 Generalization

To simplify ablation studies, we generalize (11) as follows:

$$u_k = \left( \sum_j H_j^h W_{jk}^a x_j^a \right)^{\frac{1}{a}} \quad \text{with} \quad W_{jk} = \frac{P_{jk}}{(\sum_{j'} P_{j'k})^b (\sum_{k'} P_{jk'})^{1-b}}, \tag{12}$$

where $H_j$ is given in (9)–(10), $P_{jk}$ is given in (8), and $a > 0$, $b \geq 0$, and $h \geq 0$ are the model hyper-parameters. Here, we are mainly interested in the choice $a = \frac{1}{2}$, $b = 1$, and $h = 1$, which corresponds to the original model in (11). Another special configuration is $a = 1$, $b = 0$, and $h = 0$, which corresponds to $u_k = \sum_j P_{k|j} x_j$ where $P_{k|j}$ is the conditional probability of $k$ given $j$. This configuration offers a natural benchmark for our quantum approach in that it computes $u_k$ according to classical probability theory.

### 4.4 Explainability

The contribution of the $j$-th feature to the total probability $u_k$ (local explanation) is given by the addend $H_j^h W_{jk}^a x_j^a$ in (12). Therefore, the most influential feature for the classification $k^*$ is given by $j^* = \text{argmax}_j H_j^h W_{jk^*}^a x_j^a$. In general, we use $H_j^h W_{jk^*}^a x_j^a$ to rank the features by the degree in which they contribute to the classification $k^*$.

The explanation at the class level (global explanation) is obtained by investigating the product $H_j^h W_{jk}^a$ in (12), regardless of the vector $\mathbf{x}$. The global most influential feature for each class $k$ is given by $j_k^* = \text{argmax}_j H_j^h W_{jk}^a$. In general, we use $H_j^h W_{jk}^a$ to rank the features by their global importance with respect to class $k$.

### 4.5 Computational Complexity

In the training phase, the algorithm in (11), and more generally (12), learns the joint probability $P_{jk}$ in (8) by multiplying the $J \times N$ matrix of elements $z_j^{(n)}$ with the $N \times K$ matrix of elements $y_k^{(n)}$. Entry $P_{jk}$ is given by the inner product of the $j$-th row of the left matrix (which has $N$ entries) and the $k$-th column of the right matrix (which has $N$ entries), so computing it takes time $\mathcal{O}(N)$. We do this once per element. Since the output matrix has dimension $J \times K$, there are $\mathcal{O}(JK)$ elements to consider and the total work is done in $\mathcal{O}(NJK)$. That is, the training time is at most linear in the number of samples ($N$), in the number of features ($J$), and in the number of classes ($K$).

In the prediction phase, we compute $u_k$ (and $y_k$) in (11), and more generally (12). As these are all elementwise operations on the $J \times K$ matrix of elements $P_{jk}$, the total work is done in time $\mathcal{O}(JK)$. That is, the prediction time is at most linear in the number of features ($J$), and in the number of classes ($K$), and it does not depend on the number of training samples ($N$).

Finally, we notice that the computational complexity can be further improved by using sparse matrices, and all the operations involved in the training and prediction phases can be easily parallelized (e.g., on GPUs). Thus, we expect the method to be highly scalable.

## 5   Neural Architecture

A major limitation of the algorithm presented in Section 4 is that it can be applied only when $x_j \geq 0$. Here we embed the method in a more flexible architecture that admits $x_j \in \mathbb{C}$.

Let us assume that a data instance (e.g., document) can be represented as a superposition of some hidden states $|s\rangle$ for $s = 1, ..., S$ (e.g., word embeddings). Then, we write its wave function $|\psi\rangle = \sum_s \psi_s |s\rangle$ where the coefficients $\psi_s$ generally depend on the feature vector $\mathbf{x}$ (e.g., words). We represent such coefficients with a neural network $\psi_s = \psi_s(\mathbf{x})$ that maps the feature vector $\mathbf{x} \in \mathbb{C}^J$ to the vector of wave coefficients $\psi \in \mathbb{C}^S$. Then, we write the wave function of class $k$ as $|\varphi^{(k)}\rangle = \sum_s \varphi_s^{(k)} |s\rangle$ where the coefficients $\varphi_s^{(k)}$ depend on $k$ and $s$, but not on the feature vector $\mathbf{x}$. Finally, we use the Born rule in (2) to compute the probability of $|\psi\rangle$ to collapse in $|\varphi^{(k)}\rangle$:

$$u_k = P(\psi \to \varphi^{(k)}) = |\langle \varphi^{(k)} | \psi \rangle|^2 = \left| \sum_s \bar{\varphi}_s^{(k)} \psi_s(\mathbf{x}) \right|^2. \tag{13}$$

Equation (13) is read as a neural network $\mathbf{u} = \sigma(\mathbf{\Phi v})$ where $\mathbf{\Phi}$ is a matrix of elements $\Phi_{ks} = \bar{\varphi}_s^{(k)}$, $\mathbf{v} = \psi(\mathbf{x})$ is the output of the previous layer, $\mathbf{\Phi v}$ denotes the matrix product $\sum_s \Phi_{ks} v_s$, and the activation function $\sigma(\cdot) = |\cdot|^2$ is the modulus squared. Finally, we apply a normalization layer to obtain the probabilities $y_k = u_k / \sum_k u_k \in [0, 1]$. An illustration is given in Figure 1.

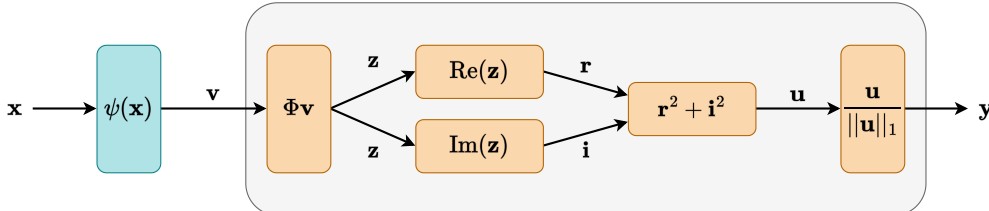

Figure 1: Born Layer (BL) architecture.

Our method is similar to the usual approach where the first part of a network is assumed to learn log-probabilities, and the final classification layer converts them into probabilities by applying the softmax function. Here, the first part of the network is assumed to learn the coefficients of a wave function, and the final classification layer converts them into probabilities by applying the Born rule.

## 5.1 Initialization

We initialize the weights $\Phi_{ks}$ such that the wave function $|\varphi^{(k)}\rangle$ has an equal probability to collapse in any state $|s\rangle$. To this end, let us write the complex-valued weights $\Phi_{ks} = \rho_{ks} e^{i\theta_{ks}}$, where $i$ is the imaginary unit, $\rho_{ks} \geq 0$, and $\theta_{ks} \in [0, 2\pi)$. We recall from (3) that the probability of $|\varphi^{(k)}\rangle$ to collapse in $|s\rangle$ is $P(\varphi^{(k)} \to s) = |\varphi_s^{(k)}|^2 = |\Phi_{ks}|^2 = \rho_{ks}^2$. We set $\rho_{ks}^2 = \rho^2$ so that the (unnormalized) probability is constant for all $k$ and $s$. To normalize the probability, we set $S\rho^2 = 1$, where $S$ is the number of states (input dimension of the layer). We obtain $\rho = 1/\sqrt{S}$, which resembles the initialization in [9] and [10]. Finally, we sample $\theta_{ks}$ from a uniform distribution in the interval $[0, 2\pi)$, such that the weights $\Phi_{ks}$ are uniformly distributed in the complex circle (isotropy).

$$\Phi_{ks} = \frac{e^{i\theta_{ks}}}{\sqrt{S}} \quad \text{with} \quad \theta_{ks} \sim \mathcal{U}(0, 2\pi). \tag{14}$$

When the feature vector $\mathbf{x}$ is a (unnormalized) probability vector as in Section 4, then (11) can be written as the neural network in (13) where we use $S = J$ and $\psi_s(\mathbf{x}) = \sqrt{x_s}$. In this case, it is interesting to initialize the weights in (13) with the corresponding weights developed in (11), that is $\Phi_{ks} = H_s \sqrt{P_{s|k}}.$[4]

## 5.2 Explainability

We notice that the probabilities $y_k$ are invariant under scaling and rotation of the coefficients $\bar{\varphi}_s^{(k)}$ in (13). To show that, we multiply $\bar{\varphi}_s^{(k)}$ by a scaling factor $\rho$ and a phase factor $e^{i\theta}$. Then, we substitute $\bar{\varphi}_s^{(k)} \to \rho e^{i\theta} \bar{\varphi}_s^{(k)}$ in (13). The phase factor $e^{i\theta}$ vanishes when computing the modulus and the scaling factor $\rho$ vanishes when the probabilities are normalized.

The invariance by scaling and rotation implies that the weights $\Phi_{ks}$ have no absolute meaning, and they become meaningful only in relation with each other. To inspect the relations among states $s$, for a given class $k$, it is interesting to visualize the weights $\Phi_{ks}$ in the complex plane (with unlabelled axes). Here, the length of the vectors can be used to rank the states by importance, while the direction of the vectors produce constructive or destructive interference among the states (see Figure 5 for an illustration). For local explanations, $\Phi_{ks} v_s$ is used instead of $\Phi_{ks}$.

---

[4]More precisely, we scale the weights by dividing them by their mean and by the square root of the number of features to mimic (14). This does not alter the model, which is invariant under scaling (see Section 5.2), but it helps to prevent vanishing or exploding gradients during backpropagation.

# 6 Empirical Results

We illustrate several aspects of our classifier using three well-established text classification benchmarks: 20Newsgroup[5], and the R8 and R52 subsets of Reuters 21578[6]. We perform tokenization using the function nltk.word_tokenize[7] and vectorize the text with TfidfVectorizer[8]. No other text transformation or cleaning procedure is performed. The final datasets are composed by (20Newsgroup) 20 classes, 204 817 words, 11 314 training documents, and 7 532 test documents; (R8) 8 classes, 33 593 words, 5 485 training documents, 2 189 test documents; (R52) 52 classes, 38 132 words, 6 532 training documents, and 2 568 test documents.

All the results are obtained using Python 3.9 on a Google Cloud Virtual Machine equipped with CentOS 7, 12 vCPU Intel Cascade Lake 85 GB RAM, 1 GPU NVIDIA Tesla A100, and CUDA 11.5.

## 6.1 Training time, prediction time, and accuracy score

We compare our methodology against a baseline of six classifiers on the 20Newsgroup dataset. With Born Classifier (BC), we refer to the algorithm presented in Section 4, where weights are computed as in equation (8) and classification probabilities are calculated as in equation (11). The baseline is composed by the algorithms Decision Tree (DT), K-Nearest Neighbors (KNN), Random Forest (RF), Support Vector Machine (SVM), Multinomial Naive Bayes (MNB), and Logistic Regression (LR). For all the algorithms in the baseline, we use the corresponding implementation in scikit-learn. All the classifiers are executed on CPU with default parameters.

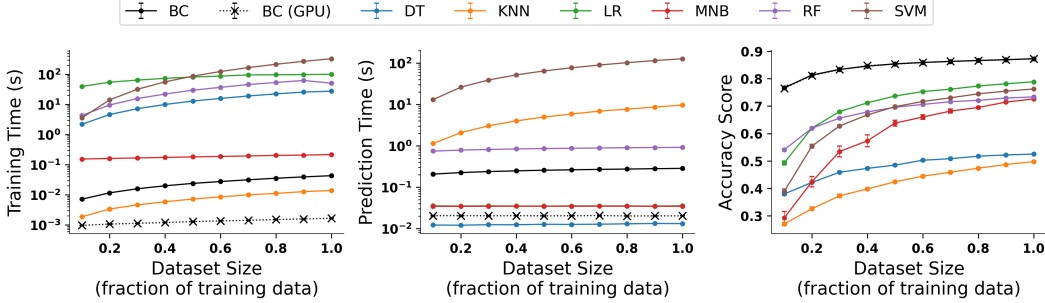

Figure 2: From left to right: training time, prediction time, and accuracy score on the 20Newsgroup dataset (y-axis) for several classifiers, in function of the fraction of data used for training (x-axis).

The comparison between BC and the baseline is reported in Figure 2, where we show the training time, prediction time, and accuracy score in function of the fraction of data used for training in 10 independent executions. Figure 2 shows that BC is fast to train, is fast to predict, can be accelerated on GPU, and it achieves the highest accuracy regardless of the size of the dataset. Moreover, as the amount of training data decreases, the accuracy gap between BC and the other classifiers widens.

## 6.2 Imbalanced data

While 20Newsgroup is almost balanced in terms of documents per class, R8 and R52 are not. In particular, the most frequent class in R52 contains 2 840 training samples, while the least frequent class contains only 1 document in the training set. Figure 3 reports the F1-macro score for BC and the baseline on the three datasets, which are increasingly imbalanced. BC outperforms the baseline models and the performance gap widens for more imbalanced data. The native capability of BC to work with imbalanced data can be traced back to (8), which computes the conditional probability of the features given the classes. Dividing the joint probability by the marginal effectively normalizes by the class imbalance.

---

[5]http://qwone.com/~jason/20Newsgroups

[6]http://archive.ics.uci.edu/ml/machine-learning-databases/reuters21578-mld

[7]See https://www.nltk.org/book/

[8]See https://scikit-learn.org/stable/modules/generated/sklearn.feature_extraction.text.TfidfVectorizer.html

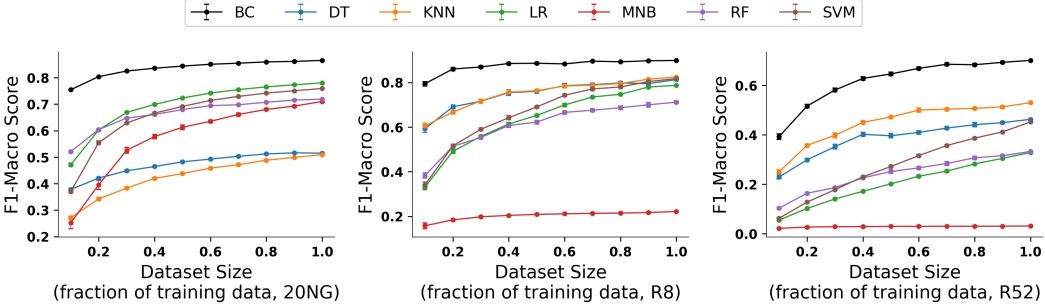

Figure 3: F1-macro score (y-axis) on 20Newsgroup, R8, R52, for several classifiers, in function of the fraction of data used for training (x-axis).

## 6.3 Hyper-parameters and ablation study

We now compare the overall accuracy score and runtime of BC with the results of the fine-tuned baseline on 20Newsgroup (see Table 1). We then conduct an ablation study on BC (see Figure 4).

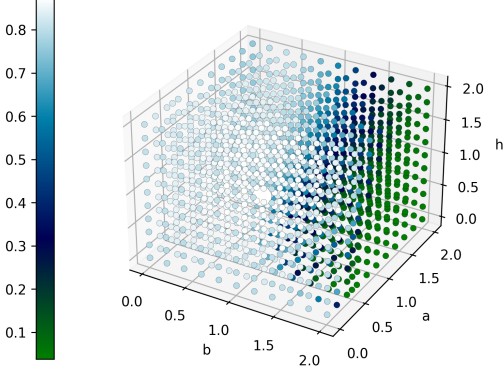

| Model | Accuracy (%) | Runtime (s) |
|---|---|---|
| DT | 53.9 | 5,583.156 |
| KNN | 55.6 | 640.574 |
| RF | 77.5 | 49,686.936 |
| SVM | 79.4 | 45,639.071 |
| LR | 82.9 | 6,066.966 |
| MNB | 84.1 | 15.320 |
| BC (11) | **87.3** | **0.043** |

Table 1: Accuracy score and runtime for several classifiers on the 20Newsgroup dataset. The runtime is the (CPU) time to optimize the model's hyper-parameters by 5-fold cross-validated grid-search on the training set, plus the time to refit the selected model. The accuracy score is the accuracy achieved by the model on the test set.

Figure 4: Ablation study on the 20Newsgroup dataset. The figure displays the test accuracy scores of the model in (12) for several values of the hyper-parameters $a$, $b$, and $h$. The biggest point identifies the configuration of hyper-parameters $a = \frac{1}{2}$, $b = 1$, $h = 1$, which corresponds to the original model in (11).

In Table 1, we tune the baseline classifiers via grid-search using 5-fold cross-validation. Except for BC, which needs no tuning, all the other classifiers use between 20 and 50 combinations of hyper-parameters (reported in the replication code). BC still achieves the highest accuracy by a remarkable margin. Furthermore, the overall time required to classify the dataset is several orders of magnitude smaller than than that required by the baseline algorithms. This is partly due to the speed of execution of BC (see Figure 2) and partly due to the fact that BC has no hyper-parameters to tune.

The ablation study is reported in Figure 4. Here we display the test accuracy scores of the model corresponding to equation (12) for several values of the hyper-parameters $a$, $b$, and $h$. The configuration $a = \frac{1}{2}$, $b = 1$, $h = 1$, corresponds to BC in equation (11) and it achieves an accuracy of 87.3%. Selecting the hyper-parameters via cross validation also achieves a test accuracy of 87.3%, suggesting that our prior generalizes well. The classical configuration $a = 1$, $b = 0$, $h = 0$, discussed in Section 4.3, achieves an accuracy of 83.3%. Removing the regularization ($h = 0$) from our model leads to a large drop in performance: from 87.3% to 77.7%. On the other hand, adding the regularization ($h = 1$) to the classical configuration does not lead to any performance gain: from 83.3% to 82.0%. These results can be understood by recalling that the classical configuration uses the conditional probability of the classes given the features ($P_{k|j}$), while BC uses the conditional

probability of the features given the classes ($P_{j|k}$). As $P_{j|k}$ gives larger weight to more frequent features, then BC is dominated by noise when the features irrelevant for classification are also the most frequent (as is typical in text). Regularization becomes important to reduce the influence of frequent features that are slightly skewed. Instead, in the classical configuration, regularization is redundant as $P_{k|j}$ already limits the influence of such features.

## 6.4 Embedding in a neural network architecture

With Born Layer (BL), we refer to the neural architecture displayed in Figure 1 that takes in input $\psi(\mathbf{x})$ and returns the classification probabilities according to equation (13). Here we choose $\psi_s(\mathbf{x}) = \sqrt{x_s}$ so that the transformation in equations (11) and (13) are the same, and they can be directly compared (in general, $\psi(\mathbf{x})$ may be an arbitrary transformation to exploit e.g., semantics, as discussed in Section 7). BL is initialized as in (14) and it is optimized on the $L^1$-loss using Adam [12] with default parameters. We also initialize BL with the weights computed by BC as discussed at the end of Section 5.1. We refer to this version with BC+BL.

We stress-test BC and BL (which only rely on word counts) against a variety of deep learning approaches reported in literature (which exploit more sophisticated representations of the document). In Table 2, we include cooperative neural networks (CoNN) [20], models that learn distributed representations of entities and documents from a knowledge base (TextEnt) [25], graph convolutional networks (TextGCN) [20], the neural attentive bag-of-entities model (NABoE) [24], and the diversified ensemble neural network (DEns) [29].

We find that BC achieves state-of-the-art accuracy on the 20Newsgroup dataset, and it is up to a million times faster than alternative approaches (0.001s). This confirms that the weights computed explicitly in (11) are a good prior. Also BL matches state-of-the-art performance, but it requires significantly longer computational times (about 34s). Initializing BL with BC requires few training epochs to further improve performance, suggesting that BC+BL can be fine-tuned relatively fast (about 3s). Finally, we notice that, while BL optimizes complex-valued weights, BC+BL uses real-valued weights. This suggests that the reason for good performance lies in the transformation itself (13), rather than in the usage of complex numbers.

| | Accuracy (%) | | | Runtime (s) | | |
|---|---|---|---|---|---|---|
| | 20NG | R8 | R52 | 20NG | R8 | R52 |
| CoNN [20] | 83.7 | N/A | N/A | 120.000 | N/A | N/A |
| TextEnt [25] | 84.5 | 96.7 | N/A | 923.089 | 556.020 | N/A |
| TextGCN [26] | 86.3 | 97.1 | 93.6 | 1206.372 | 109.184 | 186.531 |
| NABoE [24] | 86.8 | 97.1 | N/A | 152.154 | 24.110 | N/A |
| DEns [29] | 87.1 | **97.7** | 94.3 | N/A | N/A | N/A |
| BC (11) | 87.3 | 95.4 | 88.0 | **0.001** | **0.001** | **0.001** |
| BL (1 epoch) | 84.6 | 96.5 | 87.9 | 0.347 | 0.276 | 0.274 |
| BL (10 epochs) | 86.2 | 96.8 | 92.6 | 3.451 | 2.747 | 2.723 |
| BL (100 epochs) | 87.1 | 97.1 | 92.7 | 34.461 | 27.452 | 27.171 |
| BC+BL (1 epoch) | 86.9 | 97.5 | 91.8 | 0.348 | 0.278 | 0.276 |
| BC+BL (10 epochs) | **87.4** | **97.7** | **95.2** | 3.458 | 2.764 | 2.724 |
| BC+BL (100 epochs) | **87.4** | 97.2 | 94.4 | 34.521 | 27.494 | 27.124 |

Table 2: Accuracy score and runtime for several methods on the 20Newsgourp, R8, and R52 datasets. The runtime is the (GPU) time to train the model. The accuracy score is the average score achieved by the model on the test set in 10 independent runs. Standard deviations are below 0.1 (%) and are omitted from the table. The runtime for CoNN is taken from the original paper [20]. The runtime for the other methods are obtained by running the corresponding code on our hardware infrastructure.

## 6.5  Explanation

To exemplify the explanation provided by BC, we extract the global weights described in Section 4.4 and we report the 10 most influential words (out of more than 200 000) for a subset of classes in 20Newsgroup. Table 3 shows that the words do not contain stopwords or punctuation, or terms that are too general to be representative of a specific class. The ability of BC to filter out noise can be traced back to the regularization in Section 4.2 that shrinks to zero the weights of words with little discriminatory power.

| # | Baseball | Hockey | Autos | Graphics | Macintosh | Windows | Cryptography |
|---|----------|--------|-------|----------|-----------|---------|--------------|
| 1 | Phillies | NHL | car | polygon | Centris | 'AX | encryption |
| 2 | Braves | hockey | cars | TIFF | Quadra | Windows | Clipper |
| 3 | pitching | Leafs | eliot | graphics | Apple | 3.1 | clipper |
| 4 | Alomar | team | SHO | 3D | Mac | windows | crypto |
| 5 | Baseball | Devils | automotive | 3DO | Duo | W4WG | NSA |
| 6 | Players | ESPN | Callison | CView | LCIII | cica | escrow |
| 7 | Mets | Wings | Dumbest | POV | LC | font | key |
| 8 | Sox | Pens | rmt6r | cview | C650 | BJ-200 | DES |
| 9 | Cubs | playoffs | Thigpen | tdawson | BMUG | NDIS | Amanda |
| 10 | baseball | playoff | Toyota | MPEG | IIsi | Win | wiretap |

Table 3: Global explanation on 20Newsgroup as described in Section 4.4. The table shows the top 10 features (words) for the classes: Baseball (rec.sport.baseball), Hockey (rec.sport.hockey), Autos (rec.autos), Graphics (comp.graphics), Macintosh (comp.sys.mac.hardware), Windows (comp.os.ms-windows.misc), Cryptography (sci.crypt).

In Figure 5, we train BL for two epochs and we represent the evolution of the complex-valued weights described in Section 5.2. The figure represents the 10 weights that have the largest (and smallest) modulus for the class rec.sport.baseball at the end of the training. Starting from weights that are initialized randomly in all directions of the complex circle, BL learns to give larger weights to more relevant features, smaller weights to less relevant features, and to orientate the features in different directions. The weights converge in the same direction for words that are specifically related to the concept of baseball. The words hockey and NHL (National Hockey League), which are still relevant for a different class of sports, are associated with a large weight but they are also collocated in the opposite direction with respect to the baseball-related words.

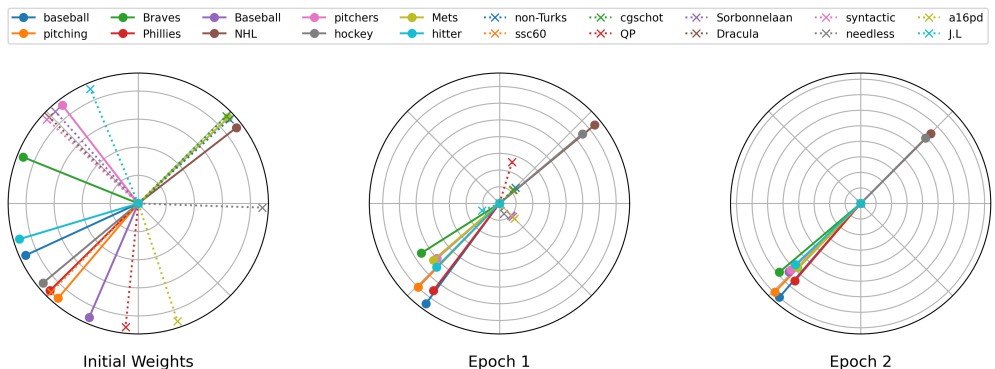

Figure 5: The figure shows the 10 most influential features (solid lines) and the 10 least influential features (dotted lines) for the class rec.sport.baseball in the 20Newsgroup dataset, as described in Section 5.2. The evolution of the (complex-valued) weights is represented from left to right.

# 7 Conclusion

By regarding text as a superposition of words, we have introduced a supervised classification algorithm based on key postulates of quantum mechanics, namely the Born rule. The classifier is self-explainable, admits a closed-form expression (11), and can be embedded in neural network architectures (13). Our method outperforms traditional algorithms on the 20Newsgroup, R8, and R52 datasets, in terms of classification performance and computational time. These results suggest that physical principles can be successfully exploited in machine learning and may open a new class of classification algorithms.

There are several potential improvements and extensions to this work. First, our classifier is derived from the Born rule and the explanation for its effectiveness in the view of machine learning remains an open question. Improving the empirical evaluation with thorough experiments would allow to assess the limitations of the method and to describe the benefits and pitfalls in various applications and data types. Second, whenever a data instance can be represented with a probability distribution over some space, then a wave function can be constructed explicitly and the data may be classified with the algorithm in Section 4. More generally, an arbitrary data instance can be classified with the network in Section 5, where the first part of the network learns the coefficients of a wave function and the final classification layer converts them into probabilities by applying the Born rule. Using the transformation in (13) on top of existing architectures offers a natural way to push our framework beyond a bag-of-words (superposition-of-words) representation of the document, to encode semantics, and to extend the method to non-textual data. For instance, an image may be classified with the network in (13) where $\psi(\mathbf{x})$ is a convolutional layer. This corresponds to regarding the image as a superposition of the features learnt by the convolution. Finally, an interesting research direction would be constructing deep networks that apply the transformation in (13) repeatedly. A deep architecture of this kind would find its biological foundations in the quantum brain hypothesis [11], which suggests that quantum events could play a non-trivial role in neuronal cells, and contribute to an extremely high complexity, variability and computational power of neuronal dynamics.

## Acknowledgments

This work was partially supported by the Google Cloud Research Credits program with the award GCP19980904.

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
