# OpenReview forum: "Text Classification with Born's Rule"
_NeurIPS.cc/2022/Conference — NeurIPS 2022 Accept_

### Official Review · Reviewer_Jcpw · 2022-07-19

**Rating:** 7
**Confidence:** 4
**Soundness:** 3 good
**Presentation:** 3 good
**Contribution:** 4 excellent

**Summary:**

The paper describes an interesting application of Born Rule to one of the most fundamental tasks in machine learning which is text classification. The Born Rule is a concept in quantum physics on estimating the probability of an object (or document in the context of this study) to collapse in one of the probable outcomes (category/class of the text). In this line, the paper frames text classification as a superposition of words where Born Rule can be integrated as a standalone classifier or with integration to a neural classifier. The authors show that the method outperform both traditional and neural network based methods on three datasets (NG, R8, and R52) by a considerable amount of margin especially with >1 epochs. There is also an emphasis on the efficient runtime obtained by using the proposed method with Born’s rule.

**Questions:**

1. What was the motivation for traditional algorithms when the use of language models such as BERT are already considered baseline in text classification as they are equally accessible compared to the tested algorithms? Also, what were the criteria for choosing the three datasets when IMDB, Amazon, Yelp, AG are often used for the evaluation of newly-proposed TC methods (see Patel et al, NAACL 2021).
2. Are there any cases where TC with Born’s rule favors particular classes based on their confusion matrices? The paper only shows Accuracy scores so there is uncertainty if it resolves class imbalance or not. This can be a limitation of the method.


**Limitations:**

The paper covers a new method for implementing text classification that seems to outperform both traditional and some neural network-based methods in terms of speed and at par with accuracy scores. The authors are advised to exhaustively list what the proposed method is (not) capable of doing covering factors such as context, how texts is processed, type of task. Again, this can be described through a table.


**Strengths And Weaknesses:**

1. The paper is fairly easy to read, and follow, and contains an arguably sufficient level of technicality that would make even intermediate readers be interested in the concepts in quantum physics such as the Born rule. I also appreciate that the authors conveniently built the method on top of Scikit and Pytorch for ease of replication on the side of the readers.
2. For this new approach, I expected a very definitive discussion on the limitations to describe the pitfalls of the proposed method in various applications and data types. Will it work on cases such as a sentiment classification task where some level of context is needed and there are cases of double negatives? Will it work on extremely noisy texts such as those collected from social media (Twitter)? What is the effect if we reduce the size of each text instance (say sentence classification instead of document-based classification)?
3. Following #2, a definitive synthesis comparing TC with Born’s rule to existing methods of text classification (using language models, traditional method) should be in order. This can be tabulated for ease of reference.

With all the strengths and weaknesses of the paper, I believe it would still be beneficial for the ML community to learn from the paper provided the authors studiously highlight the limitations of the method. Thus, I would recommend acceptance on my part.

---

> ### Author Response · Authors · 2022-07-30
> **Response to Reviewer Jcpw**
>
> - Q1: In the empirical section, our classifier relies only on a bag-of-words (superposition-of-words) representation of the document. We chose baseline models that use the same bag-of-words representaton for a fair comparison against a variety of consolidated approaches. Accordingly, we choose 20Newsgroup, R8, R52 as benchmark datasets where traditional algorithms may be expected to work reasonably well. We report Table 2 as a stress-test against approaches that use a more sophisticated representation of the document. We have edited Section 6.3 to clarify this point. The revision also improves the discussion on limitations and extensions in the conclusion.
> - Q2: BC works out-of-the-box on imabalanced data by construction. We have added Appendix A.1 in the revision, where we report F1-macro scores on the three datasets and we better discuss this point (note: we use the appendix not to violate the 9 page limit; however, we plan to move this section in the main text for the final version that allows one extra page).

---

> > ### Comment · Reviewer_Jcpw · 2022-08-03
> > **Acknowledge response from authors**
> >
> > This is to acknowledge that this reviewer has read the response of the authors. However, there are still questions in the Strengths and Weaknesses section that have not been addressed. These should also be considered.

---

> > > ### Author Response · Authors · 2022-08-06
> > > **Response to Questions in the Strengths and Weaknesses Section**
> > >
> > > We thank you for your message and we apologize for not answering those questions. We thought only the questions in the Questions section should have been addressed. We reply below to the points in the Strengths And Weaknesses section.
> > >
> > > 1. Thank you! We very much appreciate your comments on the paper and code.
> > >
> > > 2. The method itself does not encode semantics, context, or double negatives. In the empirical evaluation, we rely only on word counts (tf-idf weights). Our revision highlights that our empirical evaluation is limited to a bag-of-words representation of the document (Section 1 and 7). We don't expect BC or BL to achieve state-of-the-art results by relying only on word counts when some level of context is needed. However, we expect them to be competitive with respect to other methods that also rely on word counts. We expect BC to work out-of-the-box on extremely noisy texts thanks to the regularization proposed in Section 4.2. We also expect the method to work well if we reduce the size of each text instance, as long as the word counts are relevant for the classification task.
> > >
> > >    To better illustrate this point, we have run an additional test on the [WASSA-2017 Shared Task on Emotion Intensity (EmoInt)](http://saifmohammad.com/WebPages/EmotionIntensity-SharedTask.html). More precisely, we use the [Kaggle version](https://www.kaggle.com/datasets/anjaneyatripathi/emotion-classification-nlp) of the dataset. The dataset is almost balanced and it contains tweets classified into four emotional categories (anger, fear, joy, sadness). The task is to predict the emotional category based on the tweet's text. We choose this task because we can imagine that 1) some level of context is needed; 2) the text is noisy; 3) the size of each text instance is small. Accuracy and F1-macro scores are reported below for BC and the baseline models (we use the same tokenization and vectorization described in Section 6 for 20NG, R8, R52; the baseline is tuned via cross-validation as in Table 1 in the paper). We find that BC is competitive with the baseline models also in this scenario.
> > >
> > > |                  | **BC** | **LR** | **SVM** | **MNB** | **DT** | **RF** | **KNN** |
> > > | ---------------- | ------ | ------ | ------- | ------- | ------ | ------ | ------- |
> > > | **Accuracy (%)** | 81.8   | 80.9   | 74.9    | 71.1    | 82.0   | 78.7   | 53.2    |
> > > | **F1-macro (%)** | 81.8   | 80.9   | 74.9    | 70.8    | 78.1   | 78.3   | 52.5    |
> > >
> > > 3. Our method can be extended to encode semantics or to exploit more sophisticated representation of the document by using it on top of existing architectures (as we briefly discuss in Section 7). In this sense, our method is not an alternative to existing approaches, but it is rather a complementary resource. Although we don't have a definitive synthesis yet, we have applied preliminarily the Born layer to other approaches to see the performance improvement. Please see the response to Q2 of Reviewer mSUa.
> > >
> > > We hope that we have adequately addressed your remaining questions and we stay available for any additional clarification.

---

> > > > ### Comment · Reviewer_Jcpw · 2022-08-07
> > > > **Acknowledge additional responses from authors.**
> > > >
> > > > This is to acknowledge that I have read the authors' response. So far, my questions about the paper have been sufficiently answered and I thank the authors for taking the time to provide very detailed responses and experiments. My score will remain a 7 as I would like to see this paper at the conference.

---

### Official Review · Reviewer_mSUa · 2022-07-22

**Rating:** 5
**Confidence:** 2
**Soundness:** 3 good
**Presentation:** 2 fair
**Contribution:** 2 fair

**Summary:**

This paper proposes a novel text classification method (Born layer) based on quantum mechanics. The Born layer takes the input vector as the wave function and predicts the class of the given input with Born’s rule. The Born layer can be applied to any other neural network or function that transforms the input text into a feature vector. This paper uses the simple text transformation method, tokenization, for the transformation function, with the square root function derived by the Born rule. Experiments on three text classification datasets (20Newsgroup, R8, and R52) show the efficacy of the proposed method. The proposed method is also computationally efficient.

**Questions:**

1. Line 67: Is x_j unnormalized probability? Eq 4 states that x_j is a probability.
2. The Born layer is one of the main contributions of this paper. Do authors try any ablation study on the Born layer? e.g., apply the Born layer to other approaches and see the performance improvement.

Comments

1. In line 79, this paper describes P as the probability, but in line 83, the authors describe P as a vector. Please provide a more clear description.
2. Please provide a more description for the generalization form. What do the authors want to claim from the ablation study for the hyperparameters a and b?
3. The proposed method is applicable to other text classification tasks such as sentiment classification. More experiments on other popular text classification problems are recommended.

**Limitations:**

The authors have addressed the limitations. No negative societal impact.

**Strengths And Weaknesses:**

Strengths

1. The proposed method is novel.
2. This paper is the first trial that applies quantum theory to the text classification problem.
3. Experimental results support the validity of the proposed method.

Weaknesses

1. This paper is hard to follow. See the questions section for more details.
2. This paper does not provide any detailed description of their assumption that “text is a superposition of words.” What is the motivation that the authors see a text snippet as a superposition of words? How is this assumption brings a positive impact on text classification? What problem do existing studies have, and how does the proposed method resolve it?

---

> ### Author Response · Authors · 2022-07-30
> **Response to Reviewer mSUa**
>
> - Q1: Yes, $x_j$ is unnormalized probability. Fixed (lines 62-63)
>
> - Q2: We tried to replicate BertGCN (https://arxiv.org/abs/2105.05727) by replacing the softmax layer with BL. Classification performance improved on average for 20NG, R8, and R52 (by a limited margin). We also tried BL on non-textual classification tasks. For instance, we found that CNN + BL for image classification (MNIST, FashionMNIST, CIFAR10) slightly improves over the corresponding CNN + Softmax. Integrestingly, it seems that increasing the network depth with more BL layers widens the performance gap between BL and the corresponding architecture with Softmax. We also noticed that the networks with BL seem to require less training epochs than the corresponding architecture with Softmax to achieve optimal classification performance. Overall, these tests suggests that BL is easier to optimize. However, this point deserves thorough experimentation, while this paper mainly aims at preseenting the method. We touch upon extensions in these directions in the conclusion (lines 261-271).
> - C1: Fixed (line 82).
> - C2: In the generalized form, the hyperparameters $a=1/2$ and $b=1$ control the exponents to reproduce the transformation in Equation 6. The hyperparameter $h=1$ controls the regularization to reproduce Equation 11. In the ablation study, we show that both the transformation itself ($a$ and $b$) and the regularization ($h$) are important for performance. We also show that the classical configuration is unable to exploit the proposed regularization.
> - C3: We agree that several extensions are possible and interesting.

---

### Official Review · Reviewer_5Hcj · 2022-07-26

**Rating:** 7
**Confidence:** 2
**Soundness:** 3 good
**Presentation:** 3 good
**Contribution:** 4 excellent

**Summary:**

This paper proposes a novel algorithm derived from Born's Rule in quantum physics for text classification. The paper discusses various aspects of this classification algorithm, including training, regularization, generalization, explainability, and computational complexity. A neural architecture called Born Layer is also developed. The empirical results show the superiority of the proposed algorithm in prediction accuracy and computational complexity over several classical baseline ML algorithms. This work is promising to be the first step to develop a brand new general classification algorithm with certain explainability inspired by quantum physics.

**Questions:**

- In Figure 2, the training time of KNN increases as the dataset size grows. This implies that the compared KNN is not a brute-force one. Moreover, I would expect to see more details to be elaborated on for traditional algorithms in the supplemental material.
- The Born Classifier shows a significant superiority over other classical algorithms when the dataset size is limited. This is surprising. Do you have any interpretation of this observation? Do you have such justification for other datasets?


**Limitations:**

The author does not discuss the limitations.

**Strengths And Weaknesses:**

**Strength**

- I love the idea of a classification algorithm based on the theory of quantum physics. The authors have a comprehensive discussion of various aspects of the proposed classification algorithm.
- The algorithm is implemented with well-structured code, which should make the community easy to extend and try on more tasks.
- The algorithm shows impressive superiority over baseline algorithms in the prediction accuracy and training/inference efficiency.

**Weakness**

- The empirical result is limited. The major experiments are only conducted on 20NewsGroup. Also, the compared baselines are relatively weak. (But considering this is a preliminary work in this direction, this is acceptable.)

---

> ### Author Response · Authors · 2022-07-30
> **Response to Reviewer 5Hcj**
>
> - Q1: We have double-checked and KNN is brute-force. We have profiled the sklearn code that fits KNN. The increasing training time is due to the time to copy the input matrix. This ranges between 1 and 10 milliseconds depending on the size of the matrix. This means that training BC is almost as fast as just copying the input data.
> - Q2: We have observed this behaviour in other datasets and for different evaluation metrics. For instance, the same behaviour is observed on 20NG, R8, and R52 using F1-macro score (Figure 5 in the appendix; note: we use the appendix not to violate the 9 page limit; however, we plan to move this new section in the main text for the final version that allows one extra page). We have also observed similar behaviour on the Zoo dataset (https://archive.ics.uci.edu/ml/datasets/Zoo) and on other non-textual datasets. We conjecture that this is (at least partially) due to the regularization: the entropic re-weighting ($H_j$ in Equation 11) decreases the weights of features that are slightly skewed across classess. It seems that these features can be easily identified from fewer training samples (e.g., a stopword would appear in almost all documents and its weight is immediately shrinked to zero). As a consequence, the classification is based on less features but highly specific to a class (see e.g., Table 3). However, we plan to further investigate this point.

---

### Official Review · Reviewer_Sqxh · 2022-07-27

**Rating:** 4
**Confidence:** 5
**Soundness:** 3 good
**Presentation:** 3 good
**Contribution:** 2 fair

**Summary:**


This paper uses the mathematical tools of quantum mechanics to realize a fast and accurate text classification algorithm. Specifically, in this paper, a document is regarded as the superposition state of words, and the classification is viewed as the document to collapse to another superposition state associated with the target class. The collapse probability of the document can be computed by the Born rule. Most of the experimental results are better than baseline models. In addition, the authors also give the interpretability analysis of the two methods. The interesting experimental results show that the classification algorithm is effective and interpretable.


**Questions:**

Q1: The motivation of this paper says the prior knowledge results in more efficient algorithms. What is prior knowledge in this work, and does it give an intuitive understanding at this point?

Q2: The author says this work is the first supervised classification algorithm based on quantum theory. In my opinion, this view is not rigorous. Because many quantum-inspired models have been used to supervised classification tasks, such as QMNN[1], Complex-order[2]. In addition, there are some other quantum-inspired language approaches.e.g., quantum language models, quantum many-body wave language models. What is the advantage of this work over these language approaches?

Q3: On line 68, $|j\rangle$ is represented as word, and the document is represented as a superposition of state $|\psi\rangle = \sum_j \psi_j |j\rangle$. In quantum theory, the superposition state indicates that a quantum state is superimposed by its eigenstates, and will collapse to an eigenstate after measurement. However, the superposition state represent a document, which means that a word will be used to represent the document after measurement. Even if there is no measurement and the document is expressed as the weighted sum of word vectors in the code, its physical meaning is still unreasonable.

Q4: On line 71, author represents each class k with a wave function $|\phi^{(k)}\rangle = \sum_j \phi_j^{(k)} |j\rangle$. Why the class k is the superposition state of word? Does this mean that class k will be represented by a word in the document after measurement?

Q5: In equation 5, the $p_{j|k}$ means the conditional probability of j given k. It is necessary to specify the relationship between j and k, and the reason why the conditional probabilities are used as the superposition state coefficients.

Q6: On line 80, the $x^{(n)}$ is feature vector. Is $x^{(n)}$ a word or $|j\rangle$? What is the relationship between $y^{(n)}$ and $y_k^{(n)}$? And the $y_k^{(n)}$ has no definition.


Q7: On line 85, author says “the j-th addend does not alter the ranking of the probabilities $u_k$, thus being irrelevant for the final classification”. However, each $\sqrt{p_{j|k} x_j}$ contributes to $u_k$ in the equation 6. Why the j-th addend is irrelevant for the final classification? The reason why entropy is used as weight is not given. Can you introduce the rationality of entropy? In the equation 10, is $u_k$ a normalized vector?

Q4: On line 136, $\Phi$ is a matrix of weights， and $\Phi_{ks}$ is a scalar. Why are they equal? Is that ambiguous? Similarly, $v=v_s$ is also confusing.

Q6: Whether it can be understood in this way, the 145 to 152 line gives the initialization process of BL, and the 153 to 155 line gives the initialization process of BL by BC.

Q7: On line 159, Why does $\rho e^{i\\theta} \overline{\varphi}_s^{(k)}$ have to be derived in this form? Can't $\overline{\varphi}_s^{(k)}$ be directly interpreted?

Q8: The novelty of this work is mainly Born rule, but most quantum-inspired models use quantum measurement theory to classify or judge. What is the essential difference between this work and other quantum-inspired models?

Q9: The reasons for choosing these baseline models need to be explained. Is there comparability between this work and other baseline models in ideas, methods and other aspects? Why

[1] Li Q, Gkoumas D, Sordoni A, et al. Quantum-inspired neural network for conversational emotion recognition[C]//Proceedings of the AAAI Conference on Artificial Intelligence. 2021, 35(15): 13270-13278.

[2] Wang B, Zhao D, Lioma C, et al. Encoding word order in complex embeddings[J]. arXiv preprint arXiv:1912.12333, 2019.


**Limitations:**

The authors conclude that there are limitations that they cannot explain why quantum-inspired algorithms are better than classical algorithms. In addition, I think the limitation of this work is also the proposal of the regularization term, and whether the regularization term can be repurposed and better explained by quantum theory.


**Strengths And Weaknesses:**

From the overall perspective, the idea of this paper is interesting, and the algorithm proposed is small and exquisite. The experimental results show that the training time is greatly shortened, and the experimental accuracy is improved compared with baseline models.

However, the paper also has some weaknesses that need to be enhanced. First, the quantum-inspired models applied to supervised classification tasks is not the author's initiative. Moreover, in the experimental analysis, there is a lack of comparative experiments between this model and other quantum-inspired models. Second, the author uses quantum theory to analogize the relationship between documents and words, and the relationship between words and categories. This approach is based on intuition and has no strong theoretical basis. From the perspective of quantum measurement, this method is not reasonable. Third, the regularization lacks detailed analysis, and the use of entropy is incomprehensible. Fourth, some important information is not detailed in this paper, such as the necessity of using quantum theory, the method of quantum theory modeling prior knowledge, and what information the prior knowledge represents in the task. These makes people doubt whether it is necessary to introduce quantum theory into the problem raised in this paper.

---

> ### Author Response · Authors · 2022-07-30
> **Response to Reviewer Sqxh**
>
> - Q1: From https://en.wikipedia.org/wiki/Prior_knowledge_for_pattern_recognition
>
>   > Prior knowledge refers to all information about the problem available in addition to the training data. [...] Many classifiers incorporate the general smoothness assumption that a test pattern similar to one of the training samples tends to be assigned to the same class.
>
>   Technically, prior knowledge in this work is the invariance by scaling and rotation of the transformations that we derive (inherited by the Born rule). Qualitatively, our intuition is that quantum principles may lead to input-output relations closer to the physical/biological mechanism in which we process data/information. We discuss this point briefly in the conclusion (see lines 272-276 and reference therein) to avoid excessive speculation.
>
> - Q2: We were not aware of these works and we agree with the reviewer that "quantum theory" is therefore not rigorous. We have fixed it in the revision (lines 42-44) and we have added references to quantum language models (footnote 3). Complex-order [2] seems to be related to complex numbers but not strictly to quantum physics. QMNN [1] seems to be limited to emotion recognition, while our method is more general.
>
> - Q3:
>
>   > [...] the superposition state represent a document, which means that a word will be used to represent the document after measurement.
>
>   Correct. Physical meaning: in the analogy, this corresponds to a reader (observer) who selects (measures) a certain word (state) from a document (system).
>
> - Q4: This means that class $k$ will be represented by a word in the class after measurement. Or, better, that a reader (observer) selects (measures) a certain word from class $k$ (see above).
>
> - Q5: The identification at lines 71-72 states that the probability to observe $|j\rangle$ upon a measurement on $|\varphi^{(k)}\rangle$ is equal to the probability of the feature $j$ given the class $k$ (which is natural and can be easily computed from the matrix of co-occurences; Section 4.1). This leads to use the conditional probabilities as the superposition coefficients according to Equation 3.
>
> - Q6: The definition of $x$ and $y$ is introduced at lines 50-51. $x^{(n)}$ and $y^{(n)}$ are simply $x$ and $y$ for each training instance $n$ as defined at line 79. In the revision, we have put all vectors in bold to improve readability.
>
> - Q7:
>
>   > Why the j-th addend is irrelevant for the final classification?
>
>   Because it does not alter the ranking of classification probabiilties as explained at lines 82-85. Let $j^*$ be the "irrelevant addend" such that $P_{j^*|k}=c$ where $c$ is a constant (lines 87-88), then:
>   $$
>   k^*=\arg\max_k u_k = \arg\max_k\sqrt{u_k} = \arg\max_k \sum_j \sqrt{P_{j|k}x_j}= \arg\max_k \sum_{j\neq j^*} \sqrt{P_{j|k}x_j} + \sqrt{cx_j} = \arg\max_k \sum_{j\neq j^*} \sqrt{P_{j|k}x_j}
>   $$
>
>   > Can you introduce the rationality of entropy?
>
>   Irrelevant addends are associated to features with a uniform distribution across the classes (see above). The uniform distribution maximises entropy. Thus, we weight addends based on entropy to "shrink the contribution of irrelevant addends towards zero" (lines 85-86).
>
>   >  In the equation 10, is $u_k$ a normalized vector?
>
>   There is no $u_k$ in Equation 10. Maybe $W_{k|j}$? This is defined in Equation 9. In general, $u_k$ is unnormalized throughout the paper.
>
> - Q4bis: Fixed (lines 135-136).
>
> - Q6bis: Correct.
>
> - Q7bis: Global phases and normalization constants have no physical meaning. That's why we don't label the axes in Figure 4. The figure can be rotated and "zoomed in/out" and it would still provide the same interpretation. $\bar{\varphi}_s^{(k)}$ is interpretable only in *relative* terms.
>
> - Q8: Formalization of the Born rule as a general input-output transformation for supervised classification (lines 47-48).
>
> - Q9: In the empirical section, our classifier relies only on a bag-of-words (superposition-of-words) representation of the document. We chose baseline models that use the same bag-of-words representaton for a fair comparison against a variety of consolidated approaches. We report Table 2 as a stress-test against approaches that use a more sophisticated representation of the document. We have edited Section 6.3 to clarify this point.
>
> We hope that our response has adequately addressed your concerns. We would greatly appreaciate it if you could engage with us during the discussion period on any remaining barriers to raising your score.

---

> > ### Comment · Reviewer_Sqxh · 2022-08-03
> > **Q3 and Q8**
> >
> > For Q3, I mean that a quantum state is composed of eigenstates (or basis vectors) in superposition.  The superposition actually represents a kind of uncertainty. After quantum measurement, the quantum state collapses to a eigenstate (or a basis vector), and this eigenstate (or basis vector) is sufficient to represent the current quantum state.
> >
> > In this work, a document is composed of words in superposition. After quantum measurement, the document collapses to a word. This means the word is sufficient to represent the document. However, the semantic of the document is composed of the semantics of all the words in the document, and one word cannot represent the whole document. The amount of information is reduced from the document level to the word level by quantum measurement, which will inevitably bring information loss and semantic bias.
> >
> > For Q8, I don't think that the author has seriously addressed my question.

---

> > > ### Author Response · Authors · 2022-08-06
> > > **Q3 and Q8**
> > >
> > > Thank you for your follow up!
> > >
> > > > For Q3, I mean that a quantum state is composed of eigenstates (or basis vectors) in superposition. The superposition actually represents a kind of uncertainty. After quantum measurement, the quantum state collapses to a eigenstate (or a basis vector), and this eigenstate (or basis vector) is sufficient to represent the current quantum state. In this work, a document is composed of words in superposition. After quantum measurement, the document collapses to a word. This means the word is sufficient to represent the document.
> > >
> > > This is true at time $t=0$ when the measurement takes place. For $t>0$, the Hamiltonian will drive the evolution of the state and the word is no longer sufficient to represent the document.
> > >
> > > For example, consider a reader who reads a document. At time $t=0$ the reader reads the word $w_0$. In the analogy, this corresponds to the wave function of the document that collapses to the state $|\psi\rangle=|w_0\rangle$. Between time $t=0$ and time $t=1$, the state $|\psi\rangle$ evolves according to a Hamiltonian that maps e.g., $|\psi\rangle=|w_0\rangle$ to $|\psi\rangle=|w_1\rangle$, which represents the next word. At time $t=1$ the reader effectuates another measure on the document and reads $w_1$. The process is iterated until the reader reads the whole document.
> > >
> > > It is also possible to introduce more generic Hamiltonians. For instance, after the reader reads the word $w_0$ at time $t=0$, a Hamiltonian may map $|\psi\rangle=|w_0\rangle$ to some superposition of words that are close to $w_0$. At time $t=1$, the reader reads a random word from such superposition. This would mimic a reader who skips words and reads the document here and there.
> > >
> > > We hope this discussion clarifies the physical meaning of our method. However, we notice that the interpretation of text as a superposition of words is used in the paper to infer the wave coefficients from the distribution of words (lines 65-69). No measurement is performed and we do not make any claim about what happens to the system after measurement. This part is not needed to derive our classifier and it doesn't seem to be strictly related to the paper.
> > >
> > >
> > > > For Q8, I don't think that the author has seriously addressed my question.
> > >
> > > For Q8, the classifiers we present are general (although our empirical results are limited to text classification). The Born Classifier (BC) can be applied whenever the feature vector doesn't contain negative elements. The Born Layer (BL) can be applied whenever the input vector has elements in $\mathbb{C}$. BC is obtained explicitly by regarding a data instance as a superposition of features and by applying the Born rule. BL applies the Born rule by regarding the input vector as wave function coefficients. In this work, the Born rule is not used to mimic quantum measurement in a preparation-evolution-measurement-collapse setting (see e.g., QMNN[1]). Instead, the Born rule is viewed as introducing a new non-linear transformation from features to targets, compared to other non-linear transformations like MNB, SVM, KNN, DT, RF, etc. To our knowledge, this perspective is new and we hope that it facilitates integration of the method in mainstream ML.
> > >
> > > We hope that we have adequately addressed your remaining concerns and we stay available for any additional clarification.

---

### Author Response · Authors · 2022-07-30
**General reply to all reviewers**

We thank the reviewers for their thorough and constructive comments! Some points that relate to more than one review or that are of general interest are addressed here.

As a general remark, we would like to clarify that our method is not a novel state-of-the-art language model or a language model at all. Reviewer #5Hcj perfectly summarizes the very core of our paper and our contribution is better described as *"the first step to develop a brand new general classification algorithm with certain explainability inspired by quantum physics [...] that shows impressive superiority over baseline algorithms in the prediction accuracy and training/inference efficiency"*.

We apologize for the confusion and we have uploaded a revision that should clarify this point since the very beginning (abstract and introduction). In the revision, we also put vectors in bold to improve readability and we integrate several suggestions from the reviews. All changes are highlighted in red. Our point by point reply to individual reviews follows (note: the line numbers refer to the new revision).

We thank the reviewers for taking the time to review our work and we stay available for any additional request and clarification.

---

### Meta-Review · Area_Chair_j1PW · 2022-08-29

**Recommendation:** Accept
**Confidence:** Less certain

**Metareview:**

This  paper has 2 accepts (7) and  2 borderline accepts (5). The average is 6.
The  modification of Reviewer Sqxh does not show in the system, but he stated as follows in our discussion “I tend to modify the score of this paper to five.”


This paper shows an algorithm that delivers outstanding text classification performance despite its extreme simplicity and speed. The quantum “explanation” for the algorithm is weak. The experiments were limited to somewhat simple text classification problems, but the authors added some convincing results on sentiment analysis in their rebuttal. The paper is clearly written and the provided code is well documented, so it should be reproducible (though none of the reviewers did rerun the experiments).  The authors provided a revised version that clarified some aspects (though it did not include the additional experiments provided in the rebuttal).

The consensus after discussion is accept, given that this paper may open a new class of simple and high performance classification algorithms. The explanation for their effectiveness remains an open problem, and this paper could be improved by opening the discussion.
We recommend the following modifications in order to mitigate overreaching claims that could make the authors look naïve:

- Quantum interpretation of a sentence as superposition of words. This seems to imply that detecting a single word is sufficient to classify a sentence, and that the best solution is a combination of these weak classifiers based on a single word or N-gram. The success of Adaboost on text classification testifies of the power of these methods. However such methods do not work so well when sentences are longer, or when there is a stronger compositionality, for instance sentiment analysis (IMDB, SST). The results on EmoInt provided in the refutation are a step in the right direction. They show the method can handle some level of context, thought sentences are still short. How could this method handle long sentences? The paper would benefit from a discussion about what the superposition of words hypothesis  implies, what are it limitations, and maybe how to overcome them.

- Table 1 provides preliminary results but some of the claims can be turn off for readers with experience in linear classifiers (SVM and LR) applied to text recognition, as the authors apparently failed to select the correct algorithm (SGD classifier). Comparisons with SVMs and LR in table1 suggest BC is 1 million times faster than SVMs. However, the same difference in speed can be observed within different Sklearn SVM implementations, depending on the algorithm. The SGD classifier, which supports both SVM and LR, has a computational complexity which is even better than the O(NJK) reported in the paper, as it depends on the number of non-zero features rather than the number of features J.
Furthermore, the accuracy provided for the SVM (79.4) also seems far below the SOTA. For instance, the reference [17] reports an accuracy of 82.27 on 20NG using TF-IDF SVMs (the most vanilla setting).

- As shown in Eq.(6), the model is fundamentally linear. Explainability by taking the feature with the highest weight is as ancient as ML itself. How this method performs better than traditional linear approaches should be better illustrated. The authors just say “The words appear semantically correlated to their respective class and do not contain neither noisy words, such as stopwords or punctuation, or words whose meaning is too general to be   representative of a specific class. "

- Novelty of this paper in the field of quantum-inspired classification. Even in the revised version, while they quote work on language modeling, the authors do not seem to have quoted any work on classification, even though it was pointed by the first reviewer (https://ojs.aaai.org/index.php/AAAI/article/view/17567). As pointed in their rebuttal, this is very different from their work, but they should mention that work.



**Award:**

No

---

### Decision · Program_Chairs · 2022-09-14

Accept